# Graph-Based Retriever Captures the Long Tail of Biomedical Knowledge

Julien Delile [1]   Srayanta Mukherjee [1]   Anton Van Pamel [1]   Leonid Zhukov [1]

## Abstract

Large language models (LLMs) are transforming the way information is retrieved with vast amounts of knowledge being summarized and presented via natural language conversations. Yet, LLMs are prone to highlight the most frequently seen pieces of information from the training set and to neglect the rare ones. In biomedical research, latest discoveries are key to academic and industrial actors and are obscured by the abundance of an ever-increasing literature corpus (the information overload problem). Surfacing new associations between biomedical entities, e.g., drugs, genes, diseases, with LLMs becomes a challenge of capturing the long-tail knowledge of the biomedical scientific production. To overcome this challenge, Retrieval Augmented Generation (RAG) has been proposed to alleviate some of the shortcomings of LLMs by augmenting the prompts with context retrieved from external datasets. RAG methods typically select the context via maximum similarity search over text embeddings. In this study, we show that RAG methods may leave out a significant proportion of relevant information due to clusters of over-represented concepts in the biomedical literature. We introduce a novel information-retrieval method that leverages a knowledge graph to down-sample these clusters and mitigate the information overload problem. Its retrieval performance is about twice better than embedding similarity alternatives on both precision and recall. Finally, we demonstrate that both embedding similarity and knowledge graph retrieval methods can be combined into a hybrid model that outperforms both, enabling potential improvements to biomedical question-answering models.

[1]The Boston Consulting Group. Correspondence to: Julien Delile <julien.delile@bcg.com>.

*Accepted at the 1st Machine Learning for Life and Material Sciences Workshop at ICML 2024*. Copyright 2024 by the author(s).

## 1. Introduction

The field of biomedical research is expanding rapidly, leading to an accelerated pace of discoveries and an overwhelming surge in associated literature but contains extensive redundancies. Keeping track of the evolving landscape is thus increasingly challenging and will only be achieved with technologically advanced tools capable of filtering, summarizing, and elucidating this vast body of knowledge.

Of particular interest in the realm of summarization is Query-Based Text Summarization (QS). Unlike traditional summarization, QS tailors the summary to a user-specified question (Yu, 2022)(Yang et al., 2023). As with general text summarization (Retkowski, 2023), (Zhang et al., 2023), QS has been predominantly attempted using pre-trained models (Yu, 2022), involving zero-shot approaches (Zhang et al., 2023) and Retrieval-Augmentation Generation (RAG) (Lewis et al., 2020). QA tasks, where specific text 'chunks' need to be retrieved with high accuracy, require a broader pull of information that cover a wider spectrum of the query's nuances. While the LLM's reasoning capability and increased context length enables the ability to respond to more comprehensive queries, retrieving larger amount of information into the synthesizer context presents a distinct challenge, e.g., multi-document question-answering performance is degraded as the context grows longer (Liu et al., 2023). While new architectures designed to deal with very large context window may prevent performance drops (Yu et al., 2023), an efficient selection of the most relevant information would also reduce latency, cost, and energy consumption. This biomedical research text corpus also presents an *information overload* problem, where rare and recent yet important information is dominated by over-represented older concepts.

In this study, leaving aside the generative side of RAG, we introduce a novel knowledge-graph-based retrieval approach that enables access to the long tail of biomedical knowledge. We demonstrate that RAG retrieval approaches, leave out a significant proportion of relevant information because of the data imbalance in a queried text corpus such as Pubmed. Some over-represented topics can preclude the RAG synthesizer to access more recent discoveries by monopolizing the list of most similar text chunks. We propose to perform a rebalancing of the retrieved text chunks by under-sampling

these larger clusters of information, and to do so by structuring the text corpus with a Knowledge Graph (KG) of biomedical entities (genes, diseases and diseases). In addition, our method also provides control mechanisms to prioritize the retrieval of recent and impactful discoveries. Finally, we built a hybrid approach combining the strengths of LLM embedding semantic relationships and structured knowledge graph and show that it outperforms both embedding similarity (ES) and KG based methods for biomedical information retrieval (IR).

To the best of our knowledge, while KGs have been combined with LLMs for a variety of tasks including hallucination reduction (Ji et al., 2022)(Feng et al., 2023), LLM interpretability (Lin et al., 2019), pre-training and inference enhancement (Zhang et al., 2019)(Yasunaga et al., 2021), entity embedding (Zhang et al., 2020), link prediction (Yao et al., 2019)(Xie et al., 2022), multiple-choice QA tasks (Lin et al., 2019) (Feng et al., 2020) (Yasunaga et al., 2021) (Sun et al., 2021) (Zhang et al., 2022)), this study is the first to highlight the information overload problem in text chunk ES IR and to propose a KG IR approach to mitigate its effect.

## 2. Methods

A typical RAG workflow is composed of two sequential steps: the retrieval step and the synthesis step. We detail two alternative approaches to perform the retrieval step: i) IR using a similarity function between dense embeddings of the user question and the text corpus and ii) IR using a novel KG approach relying on entity recognition and relationship extraction (RE) performed by a model trained and fine-tuned for biomedical literature.

### 2.1. IR with text embedding similarity

#### 2.1.1. EMBEDDING INDEXING

We built an embedding index from a subset of the ~35M articles available on Pubmed (NCBI FTP site[1]). Only the article having an abstract were retained. For each experiment, a different subset was used and is specified in the associated experiment section (about 100k documents were indexed per experiment). In all cases, each selected article's title and abstract were split into individual sentences using *en_core_sci_md*, a sentence tokenizer trained on large biomedical dataset (Neumann et al., 2019).

Embeddings for each sentence were obtained using OpenAI's second generation embedding model *text-embedding-ada-002* into a 1536-dimension vector. While specifications are unknown, *text-embedding-ada-002* ranks among the top-8 retrieval text embedding models on scientific facts

benchmark (e.g. SciFact benchmark on MTEB[2]), offers a larger input size (8191 tokens) and shows clear semantic pattern on the datasets analysed in this study (Fig. 2A-B). In rare occasions ($<.01\%$), sentences longer than the input token limit were split in chunks of fixed length.

#### 2.1.2. RETRIEVAL

While complex approaches have been developed (Khattab & Zaharia, 2020), in this study we use cosine similarity to rank the embedded text chunks for each query.

### 2.2. IR with knowledge graph support

The rationale behind using a KG for IR is that traditional text ES approaches are limited by the imbalance of pieces of information in a large corpus of text such as Pubmed. While topics are often over-represented because of their importance for a field information (e.g. 84k+ EGFR references for cancer), they can hide other relevant information by their sheer number when semantic similarity is used for retrieval.

Rebalancing the retrievable text chunks can be done by undersampling the larger clusters of information. The problem then becomes how to define these clusters. While these clusters could be defined by the text chunk distributions in semantic space, biomedical literature has produced numerous ontologies covering all types of entities that can be used to organise the information. We leverage this existing knowledge to undersample clusters of information built around three types of biomedical entities (genes, diseases and chemical compounds).

#### 2.2.1. BUILDING THE BIOMEDICAL KNOWLEDGE GRAPH

To build the knowledge graph we performed 2 steps sequentially: i) The KAZU framework was used to extract entities: gene, diseases and chemical compounds(Yoon et al., 2022). It performs NER using TinyBERN2 (Sung et al., 2022) followed by entity normalization step that link single entity variations to a reference vocabulary provided by the following ontologies: Ensembl (genes), MONDO (diseases) and ChEMBL (chemical compounds). Finally, it disambiguates and/or merges overlapping candidate entities in input text chunks. ii) The PubmedBERT model (Gu et al., 2020),a BERT architecture based encoder pre-trained on Pubmed abstracts and PMC full-text articles and fine-tuned on the BioRED dataset, was used for RE from scientific abstracts(Luo et al., 2022). Pairs of disease, gene or chemical compound entities annotated by KAZU are linked in a KG if the RE model predicts a relationship between them.

---

[1]https://ftp.ncbi.nlm.nih.gov/pubmed/

[2]https://huggingface.co/spaces/mteb/leaderboard

### 2.2.2. KNOWLEDGE GRAPH INDEXING

We mapped all the text chunks produced for the embedding index onto the nodes and edges of the knowledge graph. The following rules were applied to perform the mapping: i) Only text chunks with at least one annotation are mapped ii) Text chunks with a single annotated entity are associated with the node of that entity iii) Text chunks with two annotated entities are associated with the corresponding edge if the pair has been labelled by the RE model iv) Text chunks with two annotated entities are associated with both entity nodes if the pair has not been labelled by the RE model v) Previous two rules are applied to all combinatorial entity pairs in text chunks with 3+ entities.

### 2.2.3. RETRIEVAL

Following the construction of the KG, we exploit graph distances to retrieve the chunks that are the most relevant to the user question. The first step is to identify which entity(ies) are the starting point of the graph-based retrieval. We leverage the KAZU pipeline to identify entities present in the user question. We then build the shortest path relating these entities in the graph and retrieve text chunks mapped to the shortest path entities and their neighbouring edges allowing retrieval of additional non-trivial answers. For example, to explain the relationship between two entities whose interaction is not directly documented in the literature, text chunks from neighboring entities are presented, allowing for building indirect connections.

To prioritize the most relevant text chunks that gives a fair chance to each concept mapped along the shortest path, we introduce a scoring metric that factors in both the recency and the impact of a text chunk. The impact of a text chunk is measured as the total number of citations of the associated document. Because recent articles have less citations but are more likely to contain new discoveries, we solve the trade-off between these two objectives by using the Pareto front of the recency/impact space. The combination of this ranking algorithm with the graph-distance approach is key to improving upon ES IR by rebalancing the twin objectives of impact vs recency, helping surface latest significant discoveries.

## 3. Experiments

### 3.1. Comparing KG and ES IR performance

To compare the performance of ES IR vs KG IR strategies, we purposely use an open question that requires to exploration a wide range of documents: *"What are the known drug targets for treating <disease>?"* and compared the retrieved information of both approaches with curated annotations produced by biomedical experts. We repeated the experiment over 8 diseases selected to cover the different

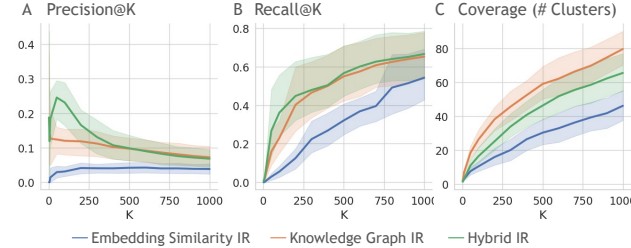

*Figure 1.* Performance comparison between ES IR, KG IR and hybrid method. Solid lines indicate the metric averages and transparent ribbon 95% confidence intervals.

therapeutic areas: asthma, pulmonary arterial hypertension, heart failure, hypertension, Parkinson's disease, Alzheimer's disease, liver cirrhosis, inflammatory bowel disease.

**Text corpus.** To ensure a broad landscape, we produced both embedding and KG indexes using Pubmed articles belonging to five therapeutic areas (nervous system, respiratory tract diseases, digestive system diseases, cardiovascular diseases, mental disorders). About 1% of the articles were sampled randomly to produce a corpus of 86166 articles and 731238 sentences. The embedding index is ordered by cosine similarity between the question embedding and the text chunk embeddings. The KG index is built and ordered by using the 372999 text chunks mapped to the graph entities or entity edges. For each question, the 1-hop neighbourhood of the question's disease entity is available for retrieval as only one entity is present in the question.

**Gold-standard dataset.** For each question, we compare the retrieved documents with a list of documents annotated by subject experts and containing both an annotation for the disease and for at least one gene in its full text. Two annotation sources were leveraged, i) a manually curated list of Medical Subject Headings (MeSH) terms for each article indexed in Pubmed/MEDLINE (Aronson et al., 2000). We selected disease-related items from their tree number (e.g. starting with *C08.127.108* for Asthma and its children diseases in MeSH). ii) We mapped the potential drug targets for Asthma using GeneRIF annotations[3]. On average, ~23 genes were identified in the subset of articles mapped to the question disease, leading to a gold-standard dataset of ~55 documents split into ~571 embeddings.

**Results.** To assess the performance of both retrieval mechanisms, we adopt two metrics widely used for IR, precision@K and recall@K, calculated by counting the number of retrieved documents, as annotations are at the document level. A document is considered retrieved if at least one of its text chunks is retrieved.

---

[3]https://www.ncbi.nlm.nih.gov/gene/about-generif

Overall, KG IR strongly outperforms ES IR on both metrics, though the precision is low in both models, mainly because the annotations are incomplete due to being voluntarily submitted. As the retrieval window increases from K=0 to K=1000, ES IR reaches a peak value of ~5% at K=250 while KG IR progressively decreases from ~12% to ~8% (Fig. 1A). Less affected by missing, yet relevant, documents in the gold-standard dataset, recall in KG IR shows a large gap of performance over ES IR (Fig. 1B). Nearly 43% of the gold-standard documents are retrieved by KG IR when K=250 text chunks are queried. In contrast, ES IR has only 17% of the gold-standard documents for the same retrieval volume. We observe ES IR recall keeps on increasing and eventually exceeds KG IR recall for a very large set of retrieved documents (K > 1000). This is because the KG IR approach only retrieves documents that have been mapped by NER to the KG while ES IR has access to the whole embedded corpus.

## 3.2. KG IR accesses the long tail of knowledge

To explain the gap in performance between both approaches, we hypothesized that KG IR is able to access the long-tail knowledge of the corpus that ES IR is missing. We investigated this hypothesis by comparing the distribution of retrieved information for both methods with the gold-standard dataset in embedding space.

**Landscape projection**. To visualise the distributions of retrieved information over the experiment text corpus, we used a non-linear dimension reduction technique, Uniform Manifold Approximation and Projection (UMAP), on all ~731k 1536-dimensional embedding vectors that learns a projection transformation that aims at maintaining the samples' local neighbourhood in low dimensional space (here 2D). Once trained and applied to the text corpus, we also transformed the embedding of one of the questions from the previous experiment (*What are the known drug targets for treating Asthma?*), allowing us to compare the question location.

**Retrieved text localisation**. To highlight the location of the retrieved text chunks in embedding space, we query the top-200 text chunks for both ES IR and KG IR methods. We apply Gaussian kernels (bandwidth factor set to 0.25) onto the UMAP coordinates of the retrieved text chunks in order to estimate the probability density function of retrieval for both methods. We then visualise the higher density regions in UMAP space by drawing filled contours for $p > 0.02$ (Fig. 2D). To assess the spread of the retrieved chunks in embedding space, we performed a k-means clustering of the ~731k chunk embeddings (k=200) and counted the number of clusters containing at least one chunk for all retrieval parameters (Fig. 1C).

**Results**. Overall, the projected embedding landscape

presents many high-density regions corresponding to over-represented concepts in the corpus (Fig. 2A-D). To assess the relevance and complexity of the landscape, we mapped the regions linked to the five covered disease areas and the various types of entities stored in the text chunks. The landscapes reveal a clear pattern of localised disease areas in embedding space (ordered from top-left to bottom-right in Fig. 2B), as well as an orthogonal pattern for the types of entities expressed in each text chunk (concentric region centred around genes top-right corner, followed by chemical compounds/drugs and diseases at the periphery Fig. 2A).

To assess whether ES IR can link the question with a diverse range of concepts, we overlayed the landscape with the cosine similarity between the question embedding and the text corpus embeddings. This reveals that most of the most similar text chunks are localized in the region surrounding the question embedding (blue cross in Fig. 2C), but also that different parts of the landscape are semantically linked to the question (black arrows in Fig. 2C). This indicates that the poor performance of ES IR is not due to its inability to build non-trivial semantic relationships but rather to access the longtail knowledge.

We hypothesized that the cause is rather the lack of data balancing that makes ES IR retrieve text chunks predominantly from the closest high-density region. This hypothesis is supported by two observations. First, comparing the region of ES IR retrieved text chunks (blue region in Fig. 2D) and from the distribution of gold-standard embedding (grey dots), we observe that the ES IR retrieval region is densely localized in the vicinity of the question embedding. In contrast, KG IR retrieval region is multipolar and covers a wider range of curated documents. A more granular comparison of the retrieved articles' text chunks in the landscape shows that KG IR also captures other smaller clusters of curated documents that were not part of the dense retrieval regions (Data not shown). Second, we quantified the spread of retrieval in embedding space by counting the number of k-means clusters that each retrieved chunk sets belong to (Fig. 2C). For the same retrieval volume, ES IR reaches less than half the number of clusters compared to KG IR. These observations lead us to the conclusion that, in contrast to ES IR, the data balancing mechanism of KG IR allows it to go beyond the immediate surrounding of the question neighborhood to retrieve relevant information thus facilitating the capture of the long-tail knowledge of biomedical information.

## 3.3. ES and KG IR are highly complementary

To combine the strengths of both methods in order to maximize the retrieval performance, we used a new ranking score averaging the min-max-normalized ES IR and KG IR scores.

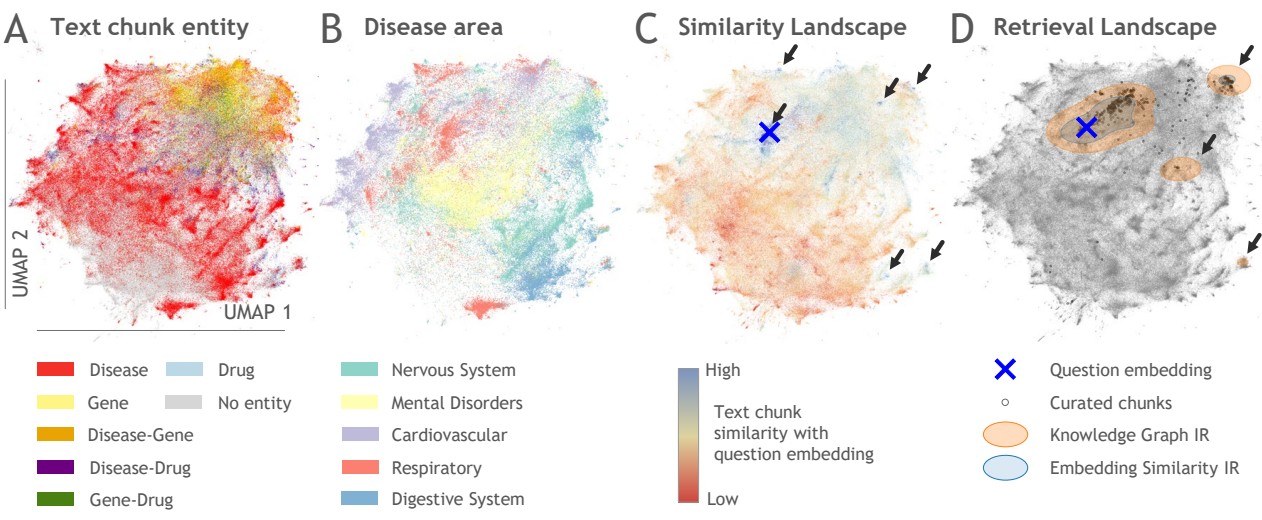

*Figure 2.* Characterization of differences between the ES IR and KG IR methods over the text embedding landscape. Each plot represents the ~731k 1536-dimensional text chunk embeddings in two dimensions via UMAP transformation.

Performing the same experiments as in 3.1., we observe that Hybrid IR strongly outperforms both ES IR and KG IR for smaller volume of retrieved information (K<100) and moderately when the retrieval window increases beyond K=250 (Fig.1A-B). Both recall and precision are about twice higher for Hybrid IR compared to KG IR for K=50. This indicates that each base retrieval method provides a complementary mechanism: data rebalancing from KG IR is not sufficient to identify the most relevant pieces of information and benefits from adding the semantic filter provided by ES.

## 4. Conclusion

To aid contextual synthesis by LLMs, IR plays a pre-eminent role to maintain a balanced and unbiased selection of retrieved information which entails extracting the long tail of biomedical information. Here, we study the strengths of a KG based information retrieval system, compare it against the more standard vector ES based retrieval system. Our findings strongly suggest that the KG based system has significantly better performance overall but is also complementary to the ES approach. Our findings further show that the tendency of ES IR to oversample the immediate neighbourhood of the question embedding due to its lack of data balancing. On the other hand, the presence of a mechanism in KG to balance the data enables the search to extend beyond the immediate neighbourhood and thereby identify a more diverse set of relevant documents. This fundamental difference between the retrieval mechanism of these two methods, also therefore spawns mutual complementarity leading to a hybrid approach being superior to both individual ones.

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
