# OpenReview forum: "Graph-Based Retriever Captures the Long Tail of Biomedical Knowledge"
_ICML.cc/2024/Workshop/ML4LMS — ML4LMS Poster_

### Official Review · Reviewer_yCVB · 2024-05-30
**The authors present a novel knowledge graph-based retrieval method to address information overload in biomedical research, significantly improving performance but with some concerns about precision and the need for varied undersampling thresholds.**

**Rating:** 9
**Confidence:** 4

**Review:**

The authors have introduced a novel information retrieval method that utilizes a knowledge graph to alleviate the problem of information overload in biomedical research. This innovative approach addresses a critical gap in the current literature by capturing the long-tail knowledge often overlooked by traditional Retrieval Augmented Generation (RAG) methods. However, I have several concerns:

1. In Figure 1A, the hybrid model, which combines embedding similarity and knowledge graph retrieval methods, outperforms both individual methods. However, there is a noticeable peak in precision for Hybrid IR from K=0 to K=250. Why does this occur? Also, in Figure 1C, the coverage of KG IR is higher than Hybrid IR. Why does this occur?

2. While the proposed method demonstrates significant improvements in retrieval performance, nearly doubling the precision and recall compared to embedding similarity alternatives, the precision remains low (less than 0.5). The authors stated that it is mainly because the annotations are incomplete. What if only use complete annotation?

3. Evaluating different undersampling thresholds (1%, 10%, 20%) may strengthen the claims of improved performance.

---

### Official Review · Reviewer_3vyu · 2024-06-10
**Review of "Graph-Based Retriever Captures the Long Tail of Biomedical Knowledge"**

**Rating:** 6
**Confidence:** 3

**Review:**

The paper presents a novel approach combining graph-based retrieval with downsampling of LLM embedding clusters to address challenges in capturing the long tail of biomedical knowledge. The methodology is intriguing and could represent a significant advancement in the field of information retrieval, particularly when integrated with generative models like RAG. However, there are a few areas where the paper could be strengthened to better support its claims and enhance its impact.

1. Definition and Clarification of Key Terms

One of the central concepts in the paper is "long tail knowledge," which is used repeatedly throughout the text. However, the term is not defined, which may lead to ambiguity about the specific challenges and goals of the proposed method.

2. Empirical Evidence on Generative Models and Long Tail Distribution

The paper posits that generative models struggle with long tail distributions but does not provide empirical evidence to support this claim. This assertion is critical as it underpins the rationale for the proposed method. It would be beneficial to include data or experiments that illustrate how generative models perform with different data distributions, particularly focusing on the long tail. This could involve comparative analysis with common benchmark tasks or a breakdown of performance metrics across frequent and rare queries.

3. Consideration of Inductive Bias in Manually Crafted Graphs

The approach relies on manually crafted graphs, which may introduce inductive biases that could skew the model's performance and limit its generalizability.

Additional Suggestions:

Comparison with Existing Methods: A comparative analysis with other IR methods, particularly those that do not use graph-based approaches, would help in delineating the benefits and potential drawbacks of the proposed method.
Broader Implications: Discussing the broader implications of the method, especially in practical biomedical applications, could enhance the paper's relevance and appeal to a wider audience.
In conclusion, while the paper proposes a promising new approach to information retrieval, addressing these points would significantly bolster the robustness of its claims and contribute to a clearer understanding of its potential impact in the field.

---

### Official Review · Reviewer_eh86 · 2024-06-12
**Interesting approach addressing important paper**

**Rating:** 8
**Confidence:** 4

**Review:**

This paper aims to illustrate how retrieval augmented generation techniques leave out relevant information due to some concepts being over-represented and introduces a new graph-based retrieval technique.

Overall, this paper is exploring a highly relevant problem at the intersection of biomedical science and LLMs. The chosen approach is supported by the literature while also being a novel solution to the identified problem. The paper is well-written, providing appropriate context on the problem and their approach.

A very minor point is that I wanted to check the `text-embedding-ada-002` on the SciFACT benchmark on MTEB and couldn't actually see it on the top 8. It looked to me as if the newer embeddings models outperformed it. This doesn't impact the validity of the results at all and it is quite possible I have misread. It could just be worth double checking!